

# Multimodal hate speech detection: a novel deep learning framework for multilingual text and images

Furqan Khan Saddozai[1], Sahar K. Badri[2], Daniyal Alghazzawi[2], Asad Khattak[3] and Muhammad Zubair Asghar[1]

[1] Gomal Research Institute of Computing, Faculty of Computing, Gomal University, D.I.Khan, KP, Pakistan
[2] Information Systems Department, Faculty of Computing and Information Technology, King Abdul Aziz University, Jeddah, Saudi Arabia
[3] College of Technological Innovation, Zayed University, Abu Dhabi Campus, Abu Dhabi, United Arab Emirates

## ABSTRACT

The rapid proliferation of social media platforms has facilitated the expression of opinions but also enabled the spread of hate speech. Detecting multimodal hate speech in low-resource multilingual contexts poses significant challenges. This study presents a deep learning framework that integrates bidirectional long short-term memory (BiLSTM) and EfficientNetB1 to classify hate speech in Urdu-English tweets, leveraging both text and image modalities. We introduce multimodal multilingual hate speech (MMHS11K), a manually annotated dataset comprising 11,000 multimodal tweets. Using an early fusion strategy, text and image features were combined for classification. Experimental results demonstrate that the BiLSTM+EfficientNetB1 model outperforms unimodal and baseline multimodal approaches, achieving an F1-score of 81.2% for Urdu tweets and 75.5% for English tweets. This research addresses critical gaps in multilingual and multimodal hate speech detection, offering a foundation for future advancements.

## INTRODUCTION

Social media platforms have revolutionized global communication but have also become significant conduits for hate speech (HS)—content targeting individuals or groups based on race, religion, or gender (*Schmid, Kümpel & Rieger, 2024*). HS fosters division and hostility (*Pálmadóttir & Kalenikova, 2018*), prompting global organizations like the United Nations to recognize it as a critical societal concern (*United Nations, 2023*). Detecting and mitigating HS in multilingual and multimodal contexts remains a pressing challenge due to the increasing prevalence of content combining text and images (*Sai, Srivastava & Sharma, 2022*) on platforms like Twitter.

Many existing HS detection methods (*Arshad et al., 2023*; *Ali et al., 2022*; *Aziz et al., 2023*) are limited to monolingual datasets and unimodal systems that analyze either text or

Corresponding author
Muhammad Zubair Asghar, mzubairgu@gmail.com

images in isolation. These methods lack in capturing the interplay between modalities that characterizes contemporary social media content. Furthermore, most prior research has focused on high-resource languages like English (*Sai, Srivastava & Sharma, 2022*; *Dwivedy & Roy, 2023*; *Gomez et al., 2020*), neglecting low-resource, linguistically diverse languages such as Urdu. The lack of resources and tailored methodologies for multilingual and multimodal contexts exacerbates these challenges.

This study proposes a novel framework combining bidirectional long short-term memory (BiLSTM) for textual feature extraction and EfficientNetB1 for visual feature processing, employing an early fusion strategy to integrate text and image modalities effectively. Additionally, we developed MMHS11K, the first multimodal, multilingual dataset comprising 11,000 annotated Urdu-English tweets, addressing critical gaps in existing resources. Experimental results demonstrate that the proposed framework outperforms state-of-the-art methods, highlighting its effectiveness in detecting HS in diverse multimodal contexts. This work contributes to the growing body of research on multimodal HS detection and sets a foundation for future exploration in low-resource multilingual contexts.

## Research gap and novelty

The growing prevalence of multimodal content on social media, particularly text and images, has necessitated advanced systems to detect HS. While many studies focus on monolingual and unimodal approaches (*Arshad et al., 2023*; *Ali et al., 2022*; *Aziz et al., 2023*), recent works (*Dwivedy & Roy, 2023*; *Gomez et al., 2020*) have explored multimodal techniques, primarily for English tweets, combining long short-term memory (LSTM) and InceptionResNetV2 models have demonstrated the efficacy of multimodal early fusion for HS detection. However, these works have been limited to monolingual datasets, often excluding low-resource languages like Urdu. The following research gaps exist in the existing literature.

1) **Lack of multimodal multilingual datasets:** Existing datasets predominantly target monolingual English content, neglecting the complexities of low-resource languages and multilingual environments like Urdu-English. Such limitations hinder the exploration of linguistic diversity in HS detection.

2) **Underutilization of advanced multimodal techniques:** While early fusion techniques have been applied, their effectiveness for multilingual multimodal content remains unexplored. No study has rigorously combined context-rich text representations with computationally efficient visual feature extraction in a low-resource multilingual setting.

To bridge this gap, a novel deep learning (DL) framework is used to detect and classify multimodal HS from multilingual (Urdu-English) text and images. In this system, the BiLSTM model is used to extract text features from preprocessed multilingual tweets. This model is able to acquire contextual information from both directions. Moreover, the

EfficientNetB1 technique is employed to extract visual features, as it requires fewer parameters while achieving high accuracy. The extracted textual and visual features are concatenated using an early fusion strategy. The combined features are used to decide the class of multimodal multilingual (Urdu-English) text and image-based tweets.

## Research questions

This study addresses:

**RQ1:** How to categorize multimodal multilingual (Urdu-English) tweets consisting of text and image into hate or no-hate classes using a novel DL model, namely BiLSTM+EfficientNetB1 with an early fusion strategy?

**RQ2:** What is the performance of the proposed model w.r.t unimodal multilingual text and image-based classification models?

**RQ3:** How to evaluate the performance of the proposed BiLSTM+EfficientNetB1 system w.r.t to baseline multimodal multilingual HS detection techniques and different datasets?

## Research contributions

This study bridges the gap in multimodal multilingual HS detection by addressing dataset scarcity, methodological limitations, and linguistic underrepresentation, marking a significant step toward inclusive and context-aware HS detection systems. The novel contributions of this work are:

1) ***Creation of the MMHS11K dataset:*** We introduce the first balanced multimodal multilingual (Urdu-English) HS detection dataset, consisting of 11,000 tweets with both textual and visual content. The dataset addresses the need for inclusive and representative resources by capturing diverse HS patterns across two languages.

2) ***Novel multimodal DL framework:*** The proposed BiLSTM+EfficientNetB1 framework leverages bidirectional contextual embeddings for text and efficient feature extraction for images. By applying an early fusion strategy, our model integrates these features to achieve superior performance compared to baseline approaches.

3) ***Comprehensive evaluation:*** Our work extends prior studies by evaluating the proposed system on multimodal multilingual data, demonstrating its robustness in detecting subtle and context-dependent HS expressions. The results highlight the effectiveness of the combined textual and visual features in improving classification accuracy.

To contextualize the contributions of this study, a detailed comparison with related methods is provided in the subsequent section (at the end of the Related Work section). This analysis highlights the limitations of prior approaches and the advancements introduced by our proposed framework.

This article is organized into different sections. The related work about HS detection is demonstrated in 'Related Work'. 'Methodology' demonstrates the methodology of the proposed system. 'Conclusions and Future Work' presents the experimental results, evaluation, limitations future work, and conclusions.

## RELATED WORK

The existing work on HS detection may be broadly categorized into two main classes, *i.e.*, unimodal techniques and multimodal techniques (*Gandhi et al., 2024*). In unimodal approaches, supervised machine learning (ML) and DL models are used to recognize and classify hate from social media posts using textual features (*Arshad et al., 2023*; *Ali et al., 2022*; *Aziz et al., 2023*; *Abro et al., 2020*). However, traditional ML algorithms need to extract handcrafted attributes or features, which is time-consuming and difficult (*Young et al., 2018*). On the other hand, DL algorithms are capable of learning better feature representations than ML techniques (*Mutanga, 2021*; *Al-Hassan & Al-Dossari, 2022*). Moreover, DL techniques are proven to outperform ML algorithms (*Bilal et al., 2023*; *Malik et al., 2024*; *Mazari, Boudoukhani & Djeffal, 2024*). The DL models use different word embedding strategies (*Bojanowski et al., 2017*) to detect and identify HS from text. However, these models investigated monolingual Urdu (*Arshad et al., 2023*; *Bilal et al., 2023*; *Rizwan, Shakeel & Karim, 2020*) or English (*Chen, McKeever & Delany, 2018*; *Zhang, Robinson & Tepper, 2019*; *Swamy, Jamatia & Gambäck, 2019*) text to detect hateful content. Another variation of unimodal techniques is to use multilingual text, such as English, Italian, Spanish, and Urdu, to detect and classify hateful content (*Rawat, Kumar & Samant, 2024*; *Mahajan, Mahajan & Kumar, 2024*).

The review of literature revealed that most of the existing systems to detect and identify HS are able to process and analyze only the textual content of the social media data (*Jahan & Oussalah, 2023*). However, multimodal systems are becoming more popular among the research community (*Dwivedy & Roy, 2023*; *Gomez et al., 2020*; *Chhabra & Vishwakarma, 2023a*). The authors are actively working on designing and developing multimodal systems to analyze both textual and visual features of online digital data to decide whether a post is hateful or not (*Rawat, Kumar & Samant, 2024*; *Chhabra & Vishwakarma, 2023a*).

Multimodal systems utilize the ability of neural models to learn feature representations from distinct modalities (*Dwivedy & Roy, 2023*). The learned feature representations from different modalities are fused together using data fusion strategies. The fused data is used to classify the multimodal content into corresponding categories. Multimodal fusion-based strategies have been applied successfully in a variety of different domains, such as multimedia event detection (*Lan et al., 2014*), image captioning (*Vinyals et al., 2015*), visual question answering (*Malinowski, Rohrbach & Fritz, 2015*), and gang violence prevention (*Blandfort et al., 2019*).

A multimodal system to detect HS from English tweets is proposed in *Gomez et al. (2020)*. A novel dataset of multimodal tweets consisting of text and images was created (MMHS150K). Pretrained convolutional neural network (Inceptionv3) and LSTM techniques were used to acquire visual and textual features, respectively. Various feature fusion techniques were used to concatenate the extracted features, such as the spatial concatenation model (SCM), textual kernels model (TKM), and feature concatenation model (FCM). The FCM technique outperformed comparing multimodal approaches and attained an F1 score of 0.704. However, there are limited tweets for the hate category.

Another system to categorize English memes into hate and no-hate classes was proposed using computer vision and natural language processing approaches (*Jadhav & Honmane, 2021*). Textual content of the memes was utilized to classify it into corresponding classes using the BiLSTM algorithm. Moreover, the convolutional neural network (CNN) algorithm was used to predict the class of images. A late fusion strategy was used to combine the individual decisions of the text and image models using the Ex-OR technique. However, they used an imbalanced dataset in their experiments.

A novel HS identification and classification system was proposed to categorize English language multimodal tweets (*Sai, Srivastava & Sharma, 2022*). The authors used the MMHS150K dataset to perform experiments. They used the Bidirectional Encoder Representations from Transformers (BERT) model to extract text features. Moreover, different pre-trained CNN architectures (such as ResNext) were used to acquire image features. In addition, they investigated various late fusion strategies. The logistic regression algorithm was used to perform classification. The results showed that the concatenation of text and image features showed strong performance than comparing unimodal systems and achieved an accuracy of 0.677. However, their results may be improved.

A framework to detect and classify multimodal tweets was proposed in *Dwivedy & Roy (2023)*. They preprocessed English tweets consisting of text and images. They used the GloVe embedding scheme to extract embeddings. The generated embeddings were forwarded to the LSTM algorithm, and textual features were extracted. For images, EfficientNetB1 and InceptionResNetV2 were used, and visual features were acquired. Finally, text and image features were concatenated and forwarded to fully connected layers, and classification was performed. The results showed that LSTM + InceptionResNetV2 outperformed the comparing baseline works. However, their work is limited to multimodal tweets in the English language.

A novel framework to detect multimodal HS from three state-of-the-art datasets was introduced using a transformer-based multilevel attention mechanism (*Chhabra & Vishwakarma, 2024a*). Their architecture consists of three distinct parts, including a combined attention-based DL mechanism, a vision attention mechanism-based encoder, and a caption attention mechanism-based encoder. The results show that the proposed framework outperformed the baselines and existing (*Chhabra & Vishwakarma, 2023b*) models. The model attained accuracies of 0.6509, 0.8790, and 0.8088 for different datasets in detecting multimodal HS.

Nowadays, social media content usually contains both images and text. Therefore, models developed to handle only textual content are not sufficient. Multimodal systems exist for languages such as English, and monolingual text and image modalities are used to decide the category of online posts. However, existing multimodal systems (*Dwivedy & Roy, 2023*; *Gomez et al., 2020*) have not considered multilingual text and images. Hence, there is a need to develop such a multimodal system for multilingual (Urdu-English) text. These systems are required to explore the performance of multimodal systems in multilingual (Urdu-English) text and image modalities. This research aims to fill this gap and develop a multimodal HS detection system for multilingual (Urdu-English) tweets.

## Comparison with related methods

The proposed BiLSTM+EfficientNetB1 framework addresses key shortcomings in existing HS detection methods. Below, we outline the limitations of related methods and demonstrate how our approach offers improvements:

**1. Shortcomings of existing methods:**

- **Monolingual focus:** Many studies (*Arshad et al., 2023*; *Gomez et al., 2020*; *Bilal et al., 2023*) have focused on detecting HS in monolingual datasets, predominantly English, while ignoring the complexities of multilingual and low-resource languages like Urdu.
- **Unimodal approaches:** Previous methods (*Arshad et al., 2023*; *Swamy, Jamatia & Gambäck, 2019*) often rely on unimodal systems (text or image), which fail to capture the contextual interplay between text and visual modalities.
- **Limited multimodal frameworks:** Although multimodal frameworks exist (*Dwivedy & Roy, 2023*; *Gomez et al., 2020*), these methods are limited to English datasets and employ traditional fusion strategies, such as late fusion, which do not fully leverage the complementary features of text and images.

**2. Strengths of the proposed framework:**

- **Multilingual multimodal dataset:** Our work introduces MMHS11K, the first balanced multimodal dataset for Urdu-English HS detection, enabling exploration in low-resource multilingual contexts.
- **Early fusion strategy:** Unlike prior methods, we utilize an early fusion approach to integrate text and image features, enhancing the model's ability to detect subtle multimodal HS patterns.
- **Superior performance:** Experimental results show that the proposed BiLSTM+EfficientNetB1 model consistently outperforms baseline unimodal and multimodal systems, achieving higher accuracy, precision, recall, and F1-scores.
- **Robustness across modalities:** By leveraging BiLSTM for text and EfficientNetB1 for image feature extraction, the framework effectively captures bidirectional contextual and visual information, ensuring robust classification.

## METHODOLOGY

This section uses different processing steps to identify HS in multimodal multilingual (Urdu-English) tweets. A novel dataset is developed to identify hateful content. The textual and visual content are preprocessed and forwarded to the feature extraction module. Multimodal HS from text and image modalities is detected using acquired features using an early fusion-based approach. Moreover, different text and image-based unimodal techniques are used to detect HS. Figure 1 presents the main modules of the proposed system. In this figure, the two labels (*i.e.*, Contribution #1 and Contribution #2) are assigned to two distinct blue-colored dotted boxes. These labels represent the main contributions or novelty of this research. Existing literature reveals that there is no multimodal dataset to detect hate from multilingual (Urdu-English) text and images.

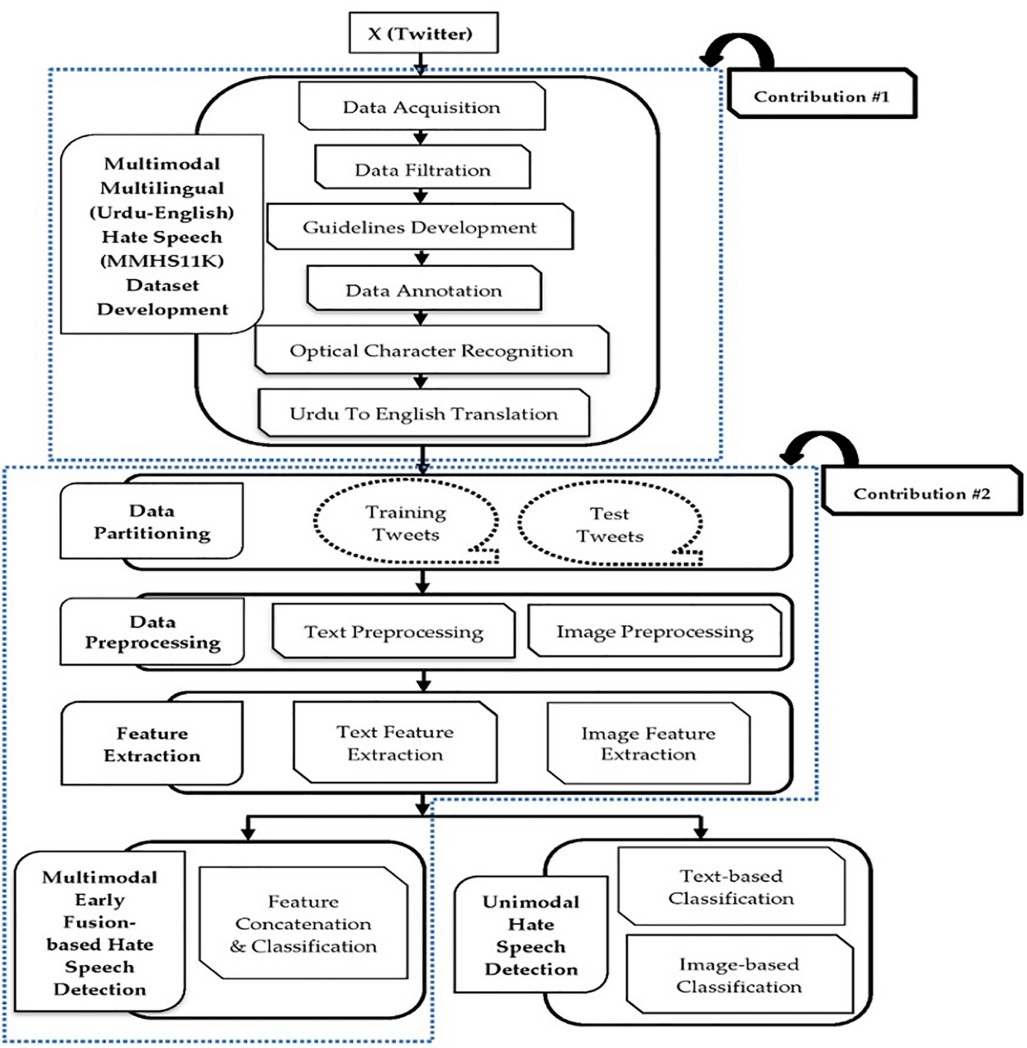

**Figure 1  Main modules of the proposed system.**

Therefore, our first contribution (Contribution #1) is to develop a novel multimodal multilingual (Urdu-English) dataset. Moreover, the existing multimodal systems have used monolingual English text and images to detect hate (*Dwivedy & Roy, 2023*; *Gomez et al., 2020*). However, these systems have not considered multilingual (Urdu-English) text and image modalities. Our proposed system aims to fill the gap by proposing a novel DL framework for detecting multimodal hate from multilingual (Urdu-English) text and images. This is the second contribution (Contribution #2) of our proposed framework. Different modules of the proposed methodology are:

- **Multimodal multilingual (Urdu-English) Hate Speech (MMHS11K) dataset development**
- **Data partitioning**
- **Data preprocessing**

**Table 1 Sample list of topics and associated keywords.**

| Topic | Keyword (Urdu) | Keyword (English) | Topic | Keyword (Urdu) | Keyword (English) |
|---|---|---|---|---|---|
| Race | پٹھان | Pathan | Sectarian | بریلوی | Barelvi |
| | سندھی | Sindhi | | دیوبندی | Deobandi |
| | پنجابی | Panjabi | | شیعہ | Shia |
| | بلوچ | Baloch | | وہابی | Wahabi |
| Politics | ڈیزل | Diesel | Institution related | جج | Judge |
| | کپتان | Captain | | فوج | Army |
| | امپورٹڈ | Imported | | جرنیل | General |
| | پٹواری | Patwari | | نیوٹرلز | Neutrals |
| National origins | پاکستان | Pakistan | Others | دہشتگرد | Terrorist |
| | بھارت | India | | کھوتا | Donkey |
| | امریکہ | America | | انتہاپسند | Extremist |
| | اسرائیل | Israel | | گیدڑ | Jackal |

- **Feature extraction**
- **Multimodal early fusion-based hate speech detection**
- **Unimodal hate speech detection**

**A. Multimodal multilingual (Urdu-English) Hate Speech (MMHS11K) dataset development**

This is the first module of our proposed multimodal HS detection system. We collect and annotate a novel dataset to identify and categorize hateful content from multimodal multilingual (Urdu-English) tweets. To create the dataset, we have developed and followed a protocol consisting of the following six phases.

**1. Data acquisition**

In this phase of our protocol, we acquire Urdu language tweets from X (https://twitter.com) as it is considered a rich source of text and image-based multimodal content. A Python script is developed to scrape tweets *via* Twitter API (*Twitter, 2022*) using the Tweepy module of Python for a period of seven months, starting from August 2022 until February 2023. The data is collected by following Twitter's terms of service. The automated script uses a number of keywords to collect tweets.

The keywords are selected from different domains such as politics, race, nationality, and sect. A sample of topics and associated keywords are presented in Table 1. We have collected multiple features of the tweets, such as tweet IDs, textual content, and corresponding images from the social media site. Tweet IDs and text are stored in a Microsoft Excel file for further processing. Meanwhile, images associated with a tweet are stored in a separate directory. If a tweet consists of more than one image, only one image is kept, and others are ignored. Finally, images are renamed with respective tweet IDs. The dataset is appended in the Supplemental Material.

**2. Data filtration**

It is important to keep useful information as unnecessary and irrelevant content may adversely affect the quality of the resultant *corpus*. Therefore, in this phase of our protocol,

| Table 2 Annotation guidelines for hate class. |
| --- |
| **Annotation guideline** |
| 1. Direct Attack: A multimodal tweets consisting of text and image that directly attacks someone due to his/her association with a specific group such as politics, race, religion *etc.* |
| 2. Hostile or threatening messages: Multimodal tweets expressing clear intentions to aggravate harm, encourage hate, or threaten some individual or groups due to their characteristics. |
| 3. Provoking law violations: Tweets encouraging someone to violate the policies or laws of government. |
| 4. Blasphemous content: Multimodal Tweets consisting of content against the glory of religion including sectarian, blasphemous and sacrilegious content. |
| 5. Linking human with animals: A multimodal tweet associating or linking human beings with animals. |
| 6. Defaming public servants: Multimodal tweets defaming public servants belonging to various departments such as army, judges, police *etc.* |
| 7. Criticizing media persons: Multimodal tweets criticizing journalists or media channels owners in an inappropriate manner for following specific agenda. |
| 8. Blaming individuals or government: Tweets blaming individuals, government or its agencies to act as foreign agents. Moreover, tweets accusing individuals or governments for taking part in national or international conspiracies. |
| 9. Spreading negativity about government: Tweets spreading or portraying negative image of government or its agencies. |
| 10. Portraying negative image of individual: Tweets portraying negative image of individual. In addition, blaming someone for being liar or coward. |
| 11. Using abusive language: Multimodal tweets containing abusive language about someone. |
| 12. Cursing people: Tweets cursing individuals, politicians or government servants. |
| 13. Sarcastic messages: A multimodal tweet provoking a sentiment of inferiority. |
| 14. Inciting violence or terrorism: Multimodal tweets that contains incitement to terrorist activities or violence. |

we have identified and removed irrelevant data using different filtration steps. We have identified and removed tweets that only contain textual content. Moreover, duplicates and tweets with non-Urdu language in text and/or images are ignored. In addition, if the visual portion of multimodal tweets consisted of only text, graphs, charts, and/or blurred images, such posts are identified and removed.

### 3. Guidelines development

This is the third phase of the *corpus* development protocol. The quality of the dataset is crucial to develop an efficient and effective classification model. To achieve this goal, we have designed guidelines for the development and annotation phases in such a way that high-quality data is obtained. We have developed a detailed set of annotation guidelines. These guidelines can also be used to develop multimodal HS datasets for languages such as Punjabi, Pashto, and Sindhi.

To develop guidelines, a subset of 2,500 multimodal tweets is selected randomly. Three experts are requested to perform manual annotation. These experts are Ph.D. scholars in computer science and native Urdu language speakers. They are asked to classify multimodal tweets consisting of text and images into Hate and No-Hate categories. Annotators have developed an initial set of annotation guidelines using their experiences during the labeling process.

These guidelines are used to compare annotated tweets, and differences are identified. We have discussed the differences and prepared a combined list of guidelines. In addition, the Fleiss Kappa measure is used to evaluate agreements. This process is iterated multiple

| Table 3 Annotation guidelines for no-hate class. |
| --- |
| **Annotation guideline** |
| 1. Expressing love or appreciation: A multimodal tweet that express love for someone is considered as no-hate. Moreover, if a tweet praises or admires someone, then it will be assigned a label of no-hate. |
| 2. News or factual information: - A multimodal tweet presenting factual information or news will be classified as no hate speech. |
| 3. Religious or intellectual quotes: If a multimodal tweet presents quotes from religious books, then it does not express hate speech. Moreover, a quote of intellectual personalities is considered as no-hate. |
| 4. Clean or hostility-free messages: Multimodal tweet that does not inflame anger or promote hostility are considered as no-hate. |
| 5. Poetry: When verse(s) of poetry are presented in multimodal tweet without any hateful expressions. Moreover, if lyrics of song are presented in a multimodal tweet. |
| 6. Conversations: - if multimodal tweet consists of conversations without using hateful expressions, then it is considered as no-hate. |
| 7. Criticism using appropriate words: - if a multimodal tweet criticizes someone using appropriate words. |
| 8. Joyful expressions: - A multimodal tweets that conveys excitement, happiness, laughter, pleasant feelings, or cheerful greeting, then it will be labeled as no-hate. |
| 9. Prayers and blessings: - A multimodal tweet expressing well-wishes and prayers for someone is considered as no-hate. |
| 10. Expressions of sorrow: - Multimodal tweets expressing sorrow without directing it to anyone. |

times to achieve a higher level of agreement among annotators. Finally, an inter-annotator agreement of 0.816 is achieved, and a refined set of guidelines is obtained. The details of the annotation guidelines are presented in Tables 2 and 3.

### 4. Data annotation

In this phase of our protocol, we asked an expert annotator to assign the label of the 'Hate' or 'No-Hate' class to each multimodal tweet using already developed guidelines. To ensure the correct application of the annotation guidelines, another expert annotator randomly verified the annotated tweets. Finally, superfluous multimodal tweets are ignored to obtain a balanced dataset.

### Optical character recognition

In this phase, the optical character recognition (OCR) technique is used to identify and extract machine-printed or handwritten text from images. In our annotated dataset, there are some multimodal tweets containing textual content in images. Hence, we use Google Tesseract (*Smith & Podobny, 2021*) to extract text from images. We have developed a Python-based script to recognize and extract Urdu text from images. The OCR text is stored in an Excel file for further processing and analysis.

### Urdu to English translation

It is the final phase of the dataset development. In this phase, tweet text (and image text if any) is translated from Urdu to English language. An automated Python script is written to translate the text using the Google Translation API.

### B. Data partitioning

In this module of the proposed system, the dataset is partitioned into two subsets. We have used a splitting ratio of 80% and 20%. The first half (*i.e.*, 80%) of the dataset is used to train the model, while the second half (*i.e.*, 20%) is used to test the performance of the

proposed system. The training and test subsets are forwarded to subsequent modules of the proposed multimodal HS detection system.

### C. Data preprocessing

In this module of the proposed system, data is preprocessed before forwarding it to subsequent modules. Preprocessing is necessary as it plays a crucial role in improving the performance of classification techniques (*Chhabra & Vishwakarma, 2024b*). Our preprocessing module works in two phases.

**Text preprocessing**

In this phase of the data preprocessing module, we have preprocessed textual content of the multimodal multilingual tweets to extract useful information from raw representations. We have developed a Python-based script to detect and remove noise from tweets (both tweet & image text). Training and test tweets are cleaned from noise (*Arshad et al., 2023*). We identify and remove all usernames (*e.g.*, @abc), URLs (*e.g.*, https://www.bbc.com/urdu), emojis (*e.g.*, 😊), extra white spaces, digits (*e.g.*, '١', '٣', '1', and '3'), diacritics (*e.g.*, " , " ), punctuations (*e.g.*, '?' , '!'), and special characters (*e.g.*, '[', '+'). Finally, tokenization is applied to both image and tweet text. We obtained tokenized representations of each corresponding tweet, *e.g.*, 'نفرت' and 'محبت'.

**Image preprocessing**

Images are considered a useful source or channel of information. It needs to be preprocessed before acquiring informative features from such a rich source of information. Therefore, in this phase of the data preprocessing module, we have applied different preprocessing strategies on visual content from both train and test sets. All images are resized to a target dimension (*i.e.*, height and width) and converted into RGB format. Moreover, image augmentation techniques such as flipping and rotation are applied to training images to increase the frequency and variance of the images. The test images are augmented using a rescaling strategy.

### D. Feature extraction

In this module of the proposed HS detection system, we extract relevant features from the entire feature space. This is necessary as feature extraction (FE) aids in dimensionality reduction and removing irrelevant features. In our proposed system, the FE module is divided into two phases to extract corresponding textual and visual features.

**Text feature extraction**

In this phase of the FE module, we extract useful features from the text modality. The preprocessed tweet and image text are used to acquire features. We have used a three-layered architecture to extract textual features.

**Embedding layer**

The traditional word representation technique, such as one-hot encoding, fails to consider the order of words and generates a huge dimensional feature; therefore, embeddings are used to overcome these issues. The embedding layer, in general, accepts sample text with n number of tokens/words. As a result, each input word is transformed into a vector of embedding $ee$. Finally, the input text is represented as a collection of

individual word embeddings such that: $= \{e_1, e_2, e_3, ......\}$. Here, $e_1$, $e_2$, and $e_3$ are the embedding vectors of the first, second, and third words. Therefore, each $e_i$ is represented as: $e_i \in \mathbb{R}^d$. For each input sample, the corresponding sentence matrix is generated as: $\mathbb{R}^{d \times n}$ such that $d$ represents the embedding dimension and $n$ is the number of words in the text sample, *i.e.*, sample length.

We have used a pre-trained word embedding model (*Haider, 2018*) to generate representations of the words. We select a subset of embeddings for each word. Each word is represented by a corresponding word vector table. The resultant vector is used to create an embedding vector of our own sequences. The extracted features of tweet and image text are forwarded to the BiLSTM layer to acquire useful feature representations.

**BiLSTM layer**

In the BiLSTM network, the input information is allowed to flow in both directions, *i.e.*, left to right and right to left (*Uysal & Gunal, 2014*). Hence, this network is able to process information from two opposing directions. It is considered a powerful technique to learn sequential dependencies between individual words and phrases from two sides of the input sequence. In other words, the BiLSTM layer adds an extra LSTM layer to reverse information flow. In our work, the BiLSTM layer processes the incoming feature vector and generates another set of features as output. The extracted feature vectors are used in subsequent layers for further processing. The working of the BiLSTM layer is presented in the following:

There are two hidden layers in the BiLSTM network (*Uysal & Gunal, 2014*). The corresponding layers are named Forward LSTM and Backward LSTM. The **Forward LSTM layer** processes the input sequence from the left to the right direction. It concatenates the past and present inputs, represented by $h_{t-1}$ and $a_t$ respectively. When an input series $a_1, a_2, a_3, ......., a_{y-1}$ is provided to this layer, the resulting sequence generated by this layer is represented by '$\overrightarrow{h}$'.

The Backward LSTM layer is responsible for processing the input sequence from the right to the left side. It concatenates the two inputs, *i.e.*, future and present input, represented by $h_{t+1}$ and $x_t$ respectively. When this layer is provided as an input sequence $a_{y+1}, ...., a_3, a_2, a_1$ the resultant sequence of this layer is represented by '$\overleftarrow{h}$'.

As a final step, the corresponding forward and backward sequences are combined, and a new review/sample matrix is produced as: $H = [h_1, h_2, h_3, .., h_c]$ such that $H \in \mathcal{R}^{c \times m}$. The concatenation or merging of both vectors is performed using Eq. (1).

$$\overleftrightarrow{h} = \overrightarrow{h} \oplus \overleftarrow{h} \tag{1}$$

The mathematical computation associated with forward LSTM (Eqs. (2)–(7)) and backward LSTM (Eqs. (8)–(13)) are presented in the following:

Forward LSTM equations:

$$i_t = \sigma(W_i[h_{t-1}, a_t] + b_i) \tag{2}$$
$$f_t = \sigma(W_f[h_{t-1}, a_t] + b_f) \tag{3}$$

$$o_t = \sigma(W_o[h_{t-1}, a_t] + b_o) \tag{4}$$
$$c\sim_t = \tau(W_c[h_{t-1}, a_t] + b_c) \tag{5}$$
$$c_t = f_t \odot c_{t-1} + i_t \odot c\sim_t \tag{6}$$
$$\rightarrow h_t = o_t \odot \tau(c_t) \tag{7}$$

Backward LSTM equations:

$$i_t = \sigma(W_i[h_{t+1}, a_t] + b_i) \tag{8}$$
$$f_t = \sigma(W_f[h_{t+1}, a_t] + b_f) \tag{9}$$
$$o_t = \sigma(W_o[h_{t+1}, a_t] + b_o) \tag{10}$$
$$c\sim_t = \tau(W_c[h_{t+1}, a_t] + b_c) \tag{11}$$
$$c_t = f_t \odot c_{t+1} + i_t \odot c\sim_t \tag{12}$$
$$\leftarrow h_t = o_t \odot \tau(c_t) \tag{13}$$

In the above equations, the terms $i_t$, $f_t$, and $o_t$ are representing various gates, *i.e.*, input, forget, and output gates, respectively. The term $\sigma$ represents a sigmoid function. $\tau$ is used to show the tangent function.

The symbol $\odot$ represents the Hadamard product. $W_i$, $W_f$, $W_c$ are representing the weight matrices associated with input, forget, and output gates. The symbols $h_{t-1}, h_{t+1}$ represent corresponding past and future states. $a_t$ is used to represent the input gate. $b_i$, $b_f, b_o, b_c$ represent the bias vectors, and $c\sim_t$ is used to represent candidate values. The symbol $c_t$ represents the state of the cell, while $c_{t-1}$, $c_{t+1}$ are used to show the past and future state of the cell.

In forward LSTM equations, the Eq. (2) represents the relevance of the current input $a_t$ and previous hidden state $h_{t-1}$. The Eq. (3) aids in discarding the parts of the previous cell state $c_{t-1}$. The output gate, represented in Eq. (4), regulates the flow of information to the hidden state.

The potential updates to the cell state are represented by Eq. (5). Equation (6) represents the updates based on input and forget gate outputs. Moreover, Eq. (7) is computed by modulating the cell state with the output gate.

In backward LSTM equations, similar computations, as in the case of forward LSTM, are performed in the backward direction. However, the next hidden state $h_{t+1}$ is used instead of the previous hidden state $h_{t-1}$.

Finally, features extracted from BiLSTM layers (both tweet and image text) are forwarded to a dense layer. Different mathematical terms used in the BiLSTM network are presented in Table 4.

**Dense layer**

A fully connected dense layer is used to accept textual features from the previous BiLSTM layer.

Furthermore, it forwards a subset of extracted text features to perform concatenation (multimodal Early fusion) or classification (unimodal). The following equation (Eq. (14)) is used to represent net input.

$$y_j = \sum x_i . w_i + b \tag{14}$$

**Table 4 Mathematical term associated with BILSTM network.**

| Mathematical terms | Definition |
| --- | --- |
| $a_t$ | Present input |
| $h_{t-1}$ | Past input |
| $h_{t+1}$ | Future input |
| $H$ | BILSTM created new review matrix |
| $\overleftrightarrow{h}$ | Final review matrix $(\overrightarrow{h_t} \oplus \overleftarrow{h_t})$ |
| $i_t, f_t, o_t$ | Input, output, and forget gate |
| $c\sim_t, c_t, c_{t-1}, c_{t+1}$ | Candidate value, cell state, past and future cell state |
| $b_i, b_f, b_o, b_c$ | Bias vectors |
| $W_i, W_f, W_o, W_c$ | Weight metrics regarding input gate, output gate, forget gate, and cell state |
| $\odot, \sigma, \tau$ | Hadamard product, tangent function, and sigmoid. |

This equation describes the linear combination of inputs, weights, and bias. In this equation, $x_i$ are the input features of the $i$-th element in the input feature vector. The $w_i$ represents the corresponding weights of the input feature. Moreover, $b$ is the bias term.

**Image feature extraction**

After extracting feature from textual content, in this phase, we have considered images to identify and acquire the visual features. A three-layered architecture is used to extract image features. The working of this phase is presented in following section.

**EfficientNetB1 layer**

We employed the EfficientNetB1 (*Tan & Le, 2019*) architecture for visual feature extraction, as it has exhibited superior performance and efficiency in various studies (*Krishna Adithya et al., 2021*; *Khan, Shahzad & Malik, 2021*). This is a model using a complex coefficient to scale all corresponding parameters by following a uniform strategy. In general, a large number of parameters are associated with pre-trained DL classifiers. Hence, these models are resource-hungry and face over-fitting problems. However, the EfficientNet classifier results in better efficiency and accuracy than ConvNet algorithms. This classifier enhances the model's performance by implementing an efficient scaling technique, thereby optimizing feature extraction and classification accuracy. The architecture of the EfficientNetB1 is presented in Fig. 2. The features extracted using the EfficientNetB1 layer are forwarded to the subsequent pooling layer for further processing.

**GlobalAvergePooling2D layer**

The GlobalAveragePooling2D layer was used to further reduce multidimensional feature vector to one dimensional vector. The working of this layer is presented in the following Eq. (15).

$$G_c = \frac{1}{H * W} \sum_{i=1}^{H} \sum_{j=1}^{W} X_{ijc} \tag{15}$$

Here, H is the height of the input feature map, W is the width of the input feature map, $X_{ijc}$ is the activation value at the position (i, j) in channel c of the input feature map.

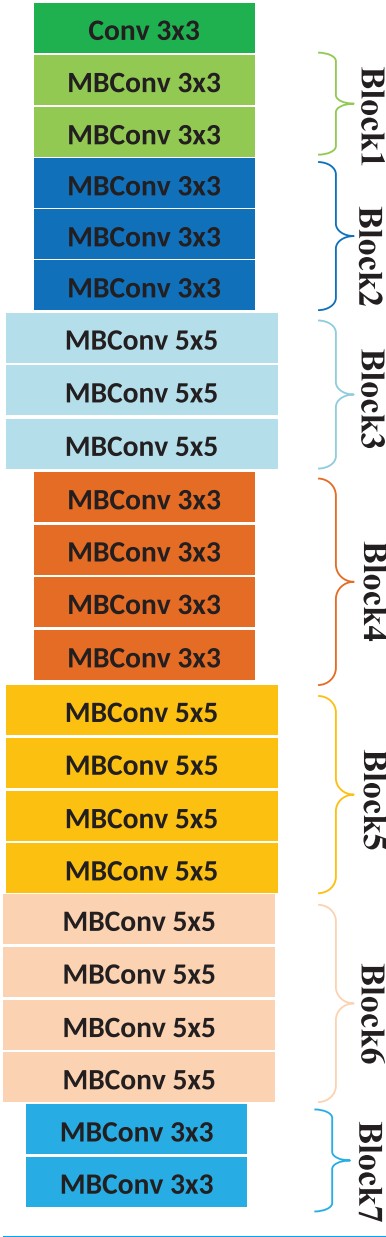

**Figure 2 Architecture of the EfficientNetB1.**

**Dense layer**

A fully connected dense layer was used to accept visual features from previous pooling layer. Furthermore, it forwarded a subset of extracted image features to perform concatenation operation (multimodal early fusion) or classification (unimodal).

**E. Multimodal early fusion-based hate speech detection**

Multimodal approaches are used to associate corresponding features from multiple different modalities. These techniques allow the model to acquire valuable information from different sources about the task at hand. Moreover, they use DL classifiers to learn and fuse the representations of data from various modalities. Data fusion is a technique to

join the features/data from different modalities Its objective is to extract relevant information from multiple sources rather than relying on a single source, thereby enhancing the performance of ML and DL classifiers. Early fusion (*Ramachandram & Taylor, 2017*), also known as feature-level fusion, is a popular data fusion strategy used to design multimodal systems. In this technique, features from multiple modalities are combined into a single feature vector. The combined feature vector is forwarded to a ML or DL classifier to train the model. Different techniques are used for joining features from different modalities, such as concatenation, product rules, and pooling (*Ramachandram & Taylor, 2017*). In this module of the proposed HS detection system, we investigate the early fusion technique to classify multimodal multilingual (Urdu-English) text and image-based tweets. We explore a concatenation strategy to detect multimodal HS. This module of the proposed system is presented in the following section.

Feature concatenation & classification

In this phase of the early HSD system, features extracted from text (both tweet and image text) and images were used to perform multimodal HSD. Various steps followed to implement early fusion strategy and final classifications of tweets are discussed.

**Feature concatenation**

We employed the concatenation technique due to its demonstrated effectiveness and robust performance in various research studies (*Sai, Srivastava & Sharma, 2022*; *Dwivedy & Roy, 2023*). Here, visual and textual features were concatenated to form a single feature set for corresponding tweets. Mathematically,

$$F_C = [F_{TT}, F_{IT}, F_I] \tag{16}$$

**Classification**

Finally, the combined features were used to classify multimodal tweets. Different layers involved in this module are:

**Dense layers**

The concatenated features from the text (both tweet and image text) are forwarded to two dense layers. These two layers are used to learn complex patterns from the input coming from previous layers.

**Dropout layer**

The outputs from previous dense layers are forwarded to the dropout layer. The drop out layer is used to avoid over-fitting.

**Final output layer**

Finally, the sigmoid function is used to classify multimodal tweets into hate or no-hate categories. This function accepts the incoming input from previous layers and calculates the probabilities of both hate and no-hate classes. The working of the proposed early fusion-based BiLSTM+EfficientNetB1 classification system is presented in Fig. 3.

**F. Unimodal hate speech detection**

In this module, the performance of text and image modalities is investigated in a unimodal fashion (*Sai, Srivastava & Sharma, 2022*). The textual and visual features in the previous FE modules are used to classify the corresponding modality into 'Hate' or 'No-Hate' categories. These techniques are investigated to show the importance of these

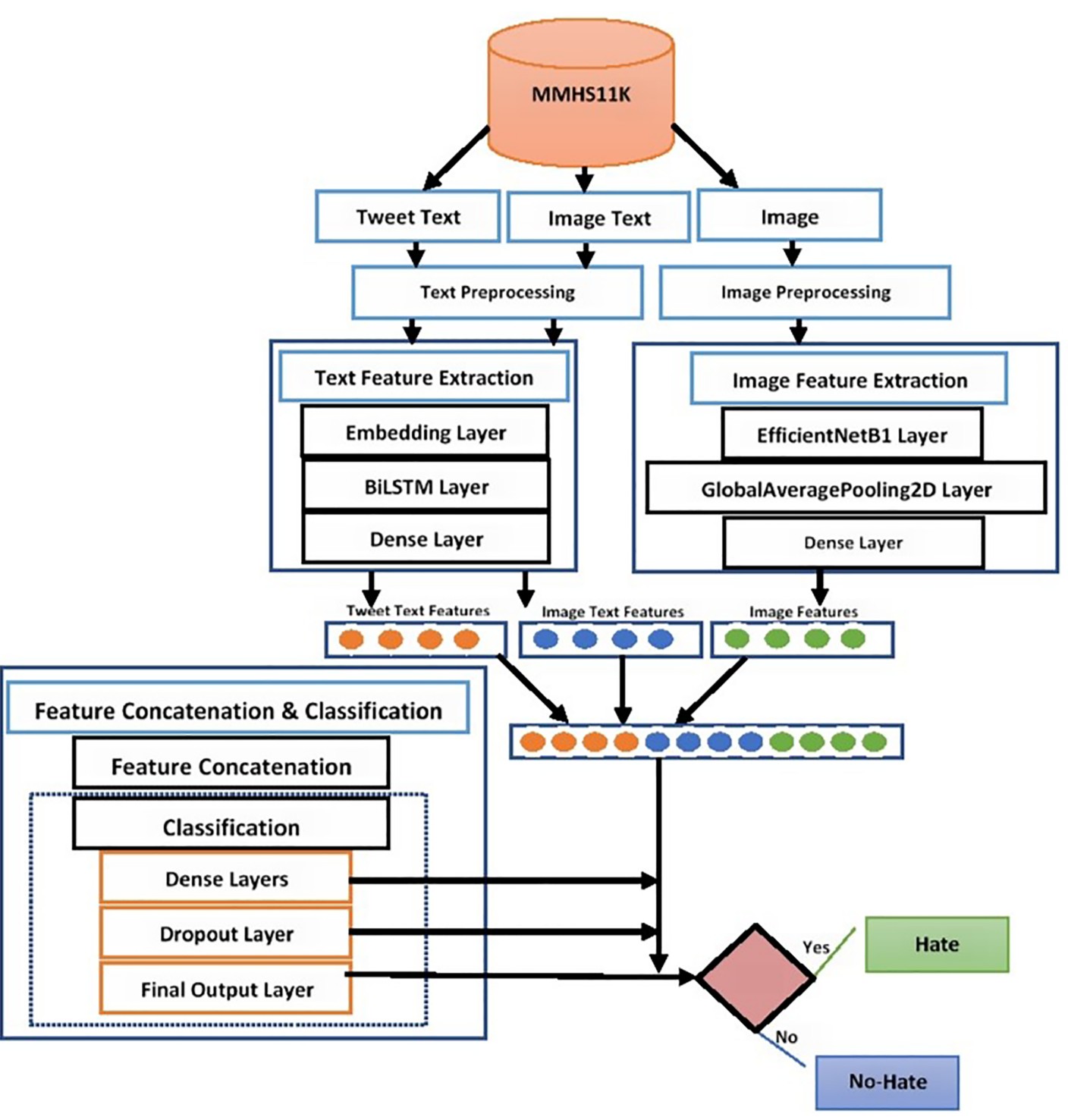

**Figure 3** Proposed early fusion-based BiLSTM+EfficientNetB1 classification system.

**Table 5 Distribution of tweets in MMHS11K.**

| Label | No. of multimodal Tweets |
|---|---|
| Hate | 5,500 |
| No-Hate | 5,500 |
| Total | 11,000 |

modalities in detecting HS. This module is further classified into (1) text-based classification, (2) image-based classification.

Text-based classification

Here, the text-based features extracted using the BiLSTM layer are used to classify text into 'Hate' and 'No-Hate' categories. Both types of text (*i.e.*, tweet text and image text) are used for classification. We forward the features to the dropout layer to avoid overfitting. The final dense layer is used to classify text into corresponding classes using the sigmoid activation function.

Image-based classification

In this unimodal image-based classification scheme, the visual features extracted from EfficientNetB1 are used to classify images into hate and no-hate categories. The acquired features are forwarded to the global average pooling layer to reduce multidimensional features to single dimensions. A dense layer is used to select a subset of features. The features are forwarded to the dropout layer to avoid overfitting. The final dense layer is used to classify images into hate or no-hate categories.

Experimental results and evaluation

In this section, details of the experimental setup and detailed answers to the corresponding research questions are presented. We have investigated the proposed multimodal technique to identify and categorize HS into corresponding categories. In addition, unimodal techniques are used to investigate the performance of the proposed system. A number of evaluation measures have been used to evaluate the performance of multimodal HS detection systems.

To perform various experiments, we have used Jupyter Notebook through the Anaconda platform. We have utilized a number of Python libraries to develop and execute computations, such as numpy, pandas, keras, and sklearn. The experiments are performed on the CPU. Python 3.11 is used as a programming language. Several routines are developed for each experiment to test and evaluate the performance of various techniques.

A. MMHS11K specifications

A number of steps are followed to generate the MMHS11K *corpus*, starting from data acquisition. In the data acquisition module, a Python-based script is used to acquire tweets from the X platform (https://twitter.com). A list of more than 400 words is used to acquire tweets. As a result, more than 200,000 tweets are obtained. The tweets are filtered using the filtration mechanism (methodology section). After development of annotation guidelines, the dataset is annotated. Furthermore, the OCR technique is used to extract text from images. Finally, the Urdu text is translated into English. The specifications of our manually

| S# | Multimodal Urdu tweet & corresponding English translation | | | Label |
|---|---|---|---|---|
| | Tweet text (Urdu) | Tweet text (English) | Associated image | |
| 2 | ! لگتاہے موٹی بڈی پھینکی گئی ہے | It seems that a thick bone has been thrown! | 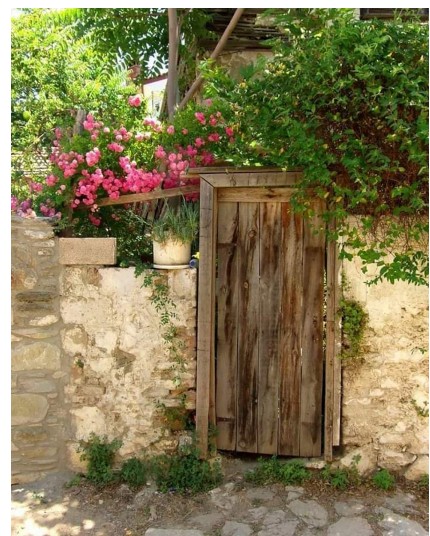 | Hate |
| 3 | پیرنی کا جادو سر چڑھ کر بول رہا ہے،ایک ایک بندہ بے بس ہو کر بغیر سوچے سمجھے بکواس کر رہا اول فول بک رہا ہے سب بے بس | The magic of Pirni is speaking loudly, each and every person is helpless and talking nonsense without thinking. | | Hate |
| 4 | گلاب جامن کے بعد پیش خدمت ہے اسٹرابری برگر | After gulab jamun is served strawberry burger | | No-Hate |
| 5 | 'صبح صبح اک خواب کی دستک پر دروازہ کھولا' دیکھا | In the morning, the door was opened on the knock of a dream | | No-Hate |
| | سرحد کے اس پار سے کچھ مہمان آئے ہیں | Some visitors have come from across the border | | |
| | آنکھوں سے مانوس تھے سارے | Everyone was familiar with the eyes | | |
| | چہرے سارے سنے سنائے | Listen to all the faces | | |
| | پاؤں دھونے، باتھ دھلانے | Wash your feet, wash your hands | | |
| | آنگن میں آسن لگوائے | Sit in the courtyard | | |
| | اور تنور پہ مکی کے کچھ موٹے موٹے روٹ پکائے | And cook some thick corn root on the oven | | |
| | پوٹلی میں مہمان مرے | Guest died in Potli | | |
| | پچھلے سالوں کی فصلوں کا گڑ لائے تھے | They brought jaggery of previous years' crops | | |

| | Table 6 (continued) | | | |
|---|---|---|---|---|
| **S#** | **Multimodal Urdu tweet & corresponding English translation** | | | **Label** |
| | **Tweet text (Urdu)** | **Tweet text (English)** | **Associated image** | |
| 6 | جب انسان اپنی غلطیوں کا خود وکیل ہو، اور دوسروں کی کوتابیوں کا خود جج ہو<br><br>تو پھر , فیصلے فاصلوں کو جنم دیتے ہیں | When a person is the lawyer of his own mistakes, and the judge of the shortcomings of others<br><br>So, decisions create distances |  | No-Hate |

developed *corpus* for detecting multimodal HS are presented in Table 5. The dataset contains 11,000 multimodal Urdu and English tweets. There are 5,500 multimodal tweets for the 'Hate' category and another 5,500 tweets for the 'No-Hate' class.

We have used an 80%, 20% ratio to divide the multimodal dataset. There are 8,800 tweets in the training set and 2,200 tweets in the test set. Each tweet consisted of a tweet's text and image portion. However, not all multimodal tweets have associated image text. There are only 2,000 tweets having associated image text in the training set. Moreover, the number of tweets with associated image text is only 500 in the test set. Samples of multimodal tweets are presented in Table 6.

B. Evaluation measures

A variety of metrics are used to test and evaluate the performance of the proposed system. Some popular evaluation metrics are accuracy, precision, recall, and F1-measure. The mathematical equations used to evaluate different metrics are presented in the following equation.

$$Accuracy = \frac{TP + TN}{TP + FP + TN + FN} \tag{17}$$

$$Precision = \frac{TP}{TP + FP} \tag{18}$$

$$Recall = \frac{TP}{TP + FN} \tag{19}$$

$$F1\text{-}score = \frac{2(Precision)(Recall)}{Precision + Recall} \tag{20}$$

**TP:** TP stands for true positive. It represents the number of samples (*i.e.*, multimodal Urdu-English tweets) which actually belong to the positive/hate class and are correctly predicted as positive.

**FP:** FP means false positive. It shows the number of samples which belong to the negative/No-Hate class. However, the classifier has wrongly predicted these samples as positive classes.

**FN:** FN represents false negative. It is used to present the number of all those samples which belong to the positive class. However, the classifier has incorrectly categorized these samples into negative classes.

**TN:** TN means true negative. TN shows the number of samples which actually belong to the negative class, and the classifier has correctly predicted these samples to the negative class.

C. Addressing first research question (RQ1)

To address the first question of our research: *How to categorize multimodal multilingual (Urdu-English) tweets consisting of text and image into hate or no-hate classes using a novel DL model, namely BiLSTM+EfficientNetB1 with early fusion strategy?* We have performed various experiments. The details of the hyperparameters are provided in Table 7. The table shows different hyperparameters investigated to classify multimodal tweets into corresponding classes.

Tweets are cleaned of noise and irrelevant material. We have used pre-trained word embeddings to acquire weights for embedding. The vocabulary (*i.e.*, Max_features) used for both tweets & image text is 5,300 words. For tweet text, we have selected 50 words from each tweet, while 45 words are selected from the image text of the corresponding tweet. The resultant embedding vectors are forwarded to two distinct BiLSTM layers for tweet and image text. We have used 150 units at the BiLSTM layer and acquired 300-dimensional feature vectors from the corresponding BiLSTM layers. Two dense layers are used to further process incoming features from previous layers. As a result, 512 textual features are obtained for both tweet text & image text.

To extract features from images, a popular CNN-based architecture is used, namely EfficientNetB1. The resized and preprocessed images are provided to EfficientNetB1. The EfficientNetB1 layer forwards its features in the shape of (7,7,1280) to the global average pooling layer (GlobalAveragePooling2D). This layer computes the average value of elements of the corresponding feature map, and a reduced spatial dimension from the corresponding feature map is obtained. All reduced dimensions are concatenated, and a one-dimensional vector is generated as a result. This vector is the summarized version of features acquired from given inputs. Therefore, we obtained reduced features in the shape of (1280,1). Finally, a dense layer is used to acquire 512 image features. These features are forwarded to the subsequent module.

Furthermore, 512 features from text (tweet and image text) and image modalities are concatenated, and a resultant 1,536-dimensional feature vector is provided to the classifier to perform classification. Finally, the 1,536 feature vector is passed through two dense layers with 1,024 and 512 neurons. Furthermore, a dropout layer is used to avoid overfitting. The final output layer with two neurons predicts probabilities of each class using a sigmoid activation function.

Using the above hyperparameters, a number of experiments are conducted to investigate the performance of the proposed system. We have used five different flavors of

Table 7 **Proposed model layers and hyper-parameters.**

| Model layers & Hyper-parameters | Hyper-parameters values |
| --- | --- |
| Tweet text | |
| Embedding layer | |
| Weights | Pre-trained word embedding |
| Max_features | 5,300 |
| Size of input vector | 50 |
| Size of output_embedding | 80 |
| Trainable | False |
| BiLSTM layer | |
| Units | 150 |
| Activation function | Relu |
| Dense layer | |
| Units | 512 |
| Activation function | Relu |
| Image text | |
| Embedding layer | Pre-trained word embedding |
| Max_features | 5,300 |
| Size of input vector | 45 |
| Size of output_embedding | 80 |
| Trainable | False |
| BiLSTM layer | |
| Units | 150 |
| Activation function | Relu |
| Dense layer | |
| Units | 512 |
| Activation function | Relu |
| Images | |
| Dimensions | 224,224,3 |
| EfficientNetB1 layer | |
| Include-top | False |
| Weights | imageNet |
| GlobalAveragePooling2D layer | |
| Dense layer | |
| Units | 512 |
| Activation function | Relu |
| Concatenation Operation: Combined 512 features from tweet text, 512 features from Image text, and 512 features from Images. A total of 1,536 features were obtained. | |
| Dense layer | |
| Units | 1,024 |
| Activation function | Relu |
| Dense layer | |
| Units | 512 |
| Activation function | Relu |

| Table 7 (continued) | |
|---|---|
| Model layers & Hyper-parameters | Hyper-parameters values |
| Dropout layer | |
| Unit size | 0.4 |
| Final output layer | |
| Units | 2 |
| Activation function | Sigmoid |
| Others | |
| Batch_size | 22, 32, 42, 64, 128 |
| Optimizer | Adam |
| Learning rate | 0.001 |
| Epochs | 20 |
| Early stopping | |
| Monitor | Val_loss |
| Patience | 2 |
| Restore best weights | True |

**Table 8 Performance of proposed BiLSTM+EfficientNetB1 system with different batch sizes.**

| S# | Model | Loss | Accuracy (%) | Precision (%) | Recall (%) | F1 (%) | batch_size |
|---|---|---|---|---|---|---|---|
| Urdu | | | | | | | |
| 1 | BiLSTM+EfficientNetB1-1 | 0.3909 | 80.1 | 79 | 78 | 78.4 | 22 |
| 2 | BiLSTM+EfficientNetB1-2 | 0.320 | 81.5 | 82.7 | 79.8 | 81.2 | 32 |
| 3 | BiLSTM+EfficientNetB1-3 | 0.403 | 79 | 77 | 75 | 76 | 42 |
| 4 | BiLSTM+EfficientNetB1-4 | 0.5 | 72 | 72 | 72 | 72 | 64 |
| 5 | BiLSTM+EfficientNetB1-5 | 0.42 | 77 | 74 | 76 | 75 | 128 |
| English | | | | | | | |
| 6 | BiLSTM+EfficientNetB1-1 | 0.56 | 65 | 65.3 | 65.1 | 65 | 22 |
| 7 | BiLSTM+EfficientNetB1-2 | 0.52 | 50 | 52 | 53 | 51 | 32 |
| 8 | BiLSTM+EfficientNetB1-3 | 0.49 | 51.2 | 51.3 | 51.2 | 52 | 42 |
| 9 | BiLSTM+EfficientNetB1-4 | 0.4 | 75.3 | 75.5 | 75.5 | 75.5 | 64 |
| 10 | BiLSTM+EfficientNetB1-5 | 0.44 | 75.1 | 75.1 | 75.1 | 75.1 | 128 |

the batch size parameter to evaluate the performance of the multimodal system. The results of various experiments are presented in Table 8. The proposed system demonstrates superior performance across multiple evaluation metrics, including accuracy, precision, recall, and F1-score, underscoring its effectiveness in hate speech detection.

The BiLSTM+EfficientNetB1 (S#:2) model with a batch size of 32, Adam optimizer, and a learning rate of 0.001 outperformed the comparing algorithm with an accuracy of 81.5% for Urdu tweets. This model showed 82.7% precision, 79.8% recall, and 81.2% F1-score. The results of corresponding models (S#:1, 3, 4, and 5) are lower (accuracy was 1.4% to 9.5% lower for different models) as compared to this model. The top-performing model shows a test loss of 0.320.

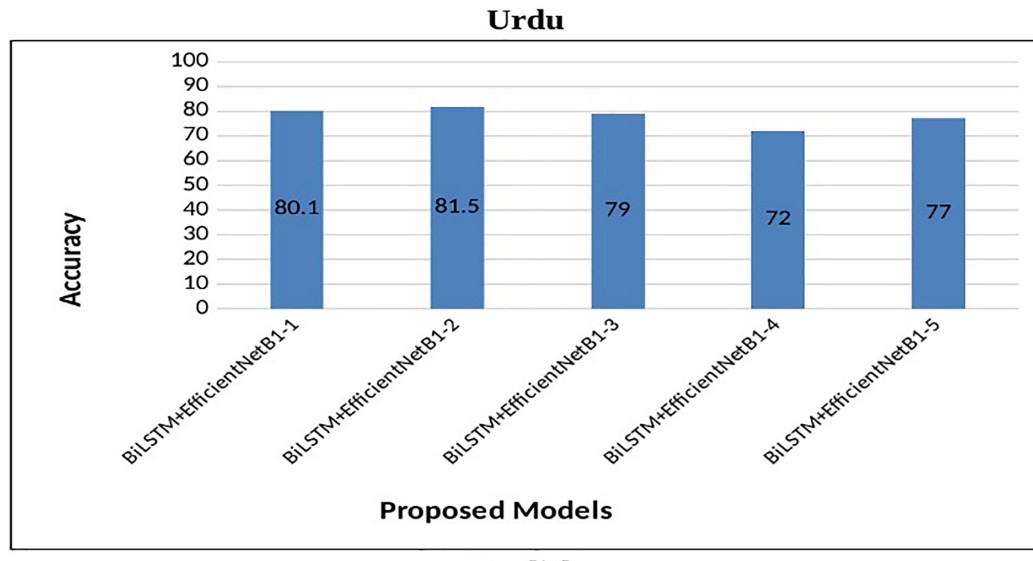

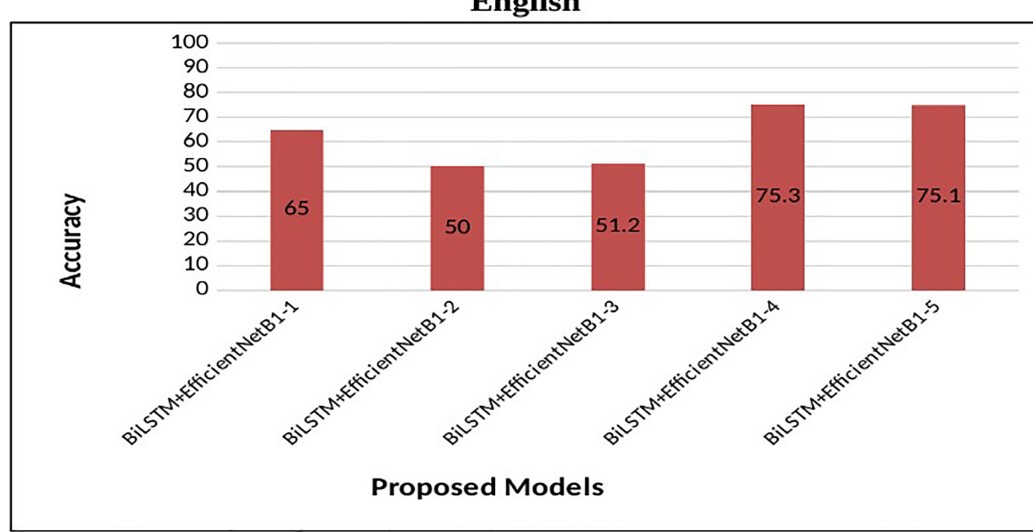

**Figure 4 Accuracy of proposed BiLSTM+EfficientNetB1 models.**

In the case of English tweets, the top-performing model (**S#:9**) attained an improved accuracy of 75.3% and a loss of 0.4 using the Adam optimizer. Figure 4 represents the performance of the proposed early fusion classifier with different parameters, as reported in Table 8. Moreover, Fig. 5 is used to show the **test loss score** of corresponding models.

**D. Addressing second research question (RQ2)**

To address the second question of our research: *What is the performance of the proposed model w.r.t unimodal multilingual text and image-based classification models?* We have performed a number of experiments. In the case of text-based (both tweet text and image text) classification, the textual features extracted from BiLSTM & dense layers are forwarded to the dropout layer to avoid over-fitting. A dense layer with two units is used to

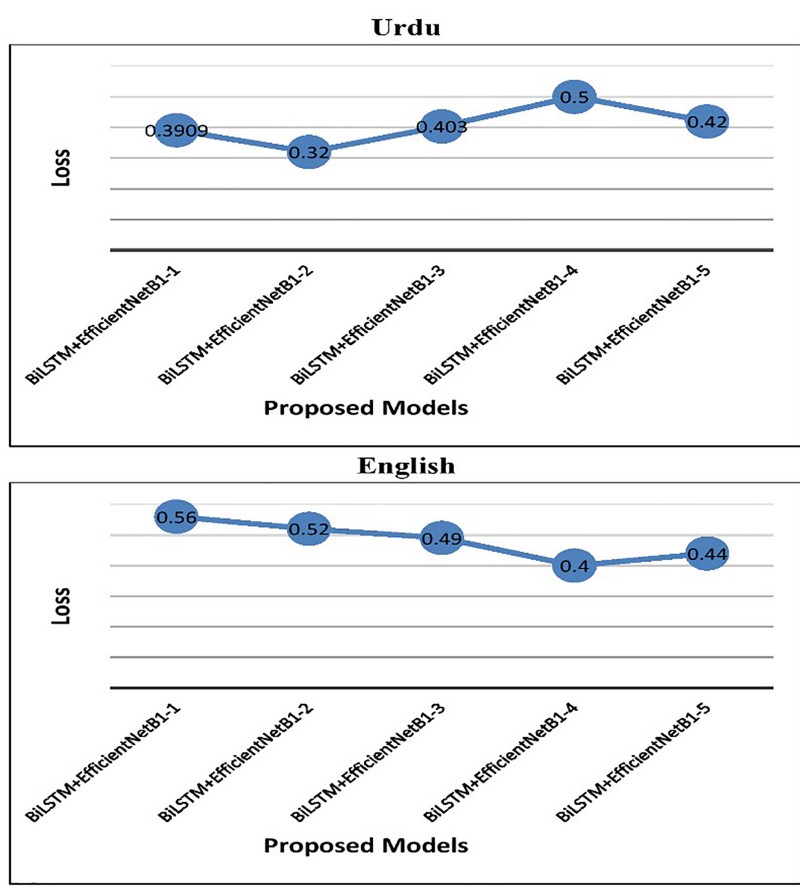

**Figure 5 Test loss of proposed BiLSTM+EfficientNetB1 models.**

classify text into 'Hate' or 'No-Hate' categories. In the case of image classification, visual features extracted from EfficientNetB1 are forwarded to the pooling layer (GlobalAveragePooling2D). This layer is used to transform multi dimensional input into one-dimensional vector. A dense layer with 512 units is used to forward the input feature to the next layer. A dropout layer is used to avoid over-fitting. Finally, a dense layer is used to classify images to the corresponding class. We have performed a series of experiments with text and image classification algorithms using above mentioned hyper-parameters. We have used Adam optimizer and learning rate of 0.001 for performing experiments. The results of various experiments for tweet text-based models are presented in Table 9. The above table shows that BiLSTM (S#:2) outperformed corresponding models with an accuracy of 77.2%, precision of 77.1%, recall of 77.4% and F1-score of 77.3%. This model used a batch size of 32 for Urdu tweets. Another BiLSTM (S#:3) showed almost similar results with an accuracy of 77% using a batch size of 42. In the case of English tweets, the BiLSTM model (S#:8) reached an accuracy of 70.5%.

In the case of image text, the performance of classification algorithms is reported in Table 10. It is obvious that the BiLSTM model (S#:1) is the top performing model among corresponding image text-based models. It has attained an accuracy of 51.6%. For

**Table 9 Performance of tweet text-based models.**

| S# | Model | Loss | Accuracy (%) | Precision (%) | Recall (%) | F1 (%) | batch_size |
|---|---|---|---|---|---|---|---|
| Urdu | | | | | | | |
| 1 | BiLSTM-1 | 0.507 | 74.4 | 77.1 | 69.3 | 73 | 22 |
| 2 | BiLSTM-2 | 0.42 | 77.2 | 77.1 | 77.4 | 77.3 | 32 |
| 3 | BiLSTM-3 | 0.483 | 77 | 79.5 | 72.6 | 75.9 | 42 |
| 4 | BiLSTM-4 | 0.505 | 76 | 75 | 78 | 76.5 | 64 |
| 5 | BiLSTM-5 | 0.553 | 55.4 | 53.4 | 84.1 | 65.3 | 128 |
| English | | | | | | | |
| 6 | BiLSTM-1 | 0.65 | 72 | 72 | 72 | 72 | 22 |
| 7 | BiLSTM-2 | 0.59 | 70 | 70.1 | 70.1 | 70 | 32 |
| 8 | BiLSTM-3 | 0.61 | 70.5 | 70.2 | 70.3 | 70.1 | 42 |
| 9 | BiLSTM-4 | 0.591 | 69 | 69 | 69 | 69 | 64 |
| 10 | BiLSTM-5 | 0.613 | 67 | 67.1 | 67.1 | 67.4 | 128 |

**Table 10 Performance of image text-based models.**

| S# | Model | Loss | Accuracy (%) | Precision (%) | Recall (%) | F1 (%) | batch_size |
|---|---|---|---|---|---|---|---|
| Urdu | | | | | | | |
| 1 | BiLSTM-1 | 0.692 | 51.6 | 50.8 | 95.8 | 66.4 | 22 |
| 2 | BiLSTM-2 | 0.694 | 50 | 50 | 100 | 66.6 | 32 |
| 3 | BiLSTM-3 | 0.692 | 49.9 | 47.5 | 1.7 | 3.3 | 42 |
| 4 | BiLSTM-4 | 0.696 | 50 | 50 | 100 | 66.6 | 64 |
| 5 | BiLSTM-5 | 0.693 | 50.5 | 50.2 | 94.8 | 65.7 | 128 |
| English | | | | | | | |
| 6 | BiLSTM-1 | 0.71 | 40 | 40 | 40 | 40 | 22 |
| 7 | BiLSTM-2 | 0.7 | 45 | 45 | 45 | 45 | 32 |
| 8 | BiLSTM-3 | 0.75 | 47.6 | 47.6 | 47.6 | 47.6 | 42 |
| 9 | BiLSTM-4 | 0.69 | 30 | 30 | 30 | 30 | 64 |
| 10 | BiLSTM-5 | 0.74 | 43 | 42.5 | 42.4 | 43 | 128 |

English-based image text, the BiLSTM model (S#:8) reached an accuracy of 47.6%. However, these accuracy values are much lower than those of tweet text models. There can be a variety of reasons for low performance. Firstly, there are only 2,000 training samples (image text) in the training set. Hence, the classifier is not properly trained to learn how to identify 'Hate' and 'No-Hate' text.

Secondly, the quality of text extracted from the image is poor. It contained a lot of noise. Hence, the results are poor in the case of image text.

Thirdly, deep learning (DL) models require large amounts of data and demonstrate superior performance when trained on extensive datasets.

For image-based classification, the results of different classification algorithms are presented in Table 11. The results show that EfficientNetB1 (S#:1) is top-performing

**Table 11 Performance of different image-based classification models.**

| S# | Model | Loss | Accuracy (%) | Precision (%) | Recall (%) | F1 (%) | batch_size |
|----|-------|------|--------------|---------------|------------|--------|------------|
| 1 | EfficientNetB1-1 | 0.48 | 69 | 69 | 68 | 68.4 | 22 |
| 2 | EfficientNetB1-2 | 0.53 | 68.5 | 68.4 | 68.5 | 68.4 | 32 |
| 3 | EfficientNetB1-3 | 0.54 | 68 | 69 | 67 | 67.9 | 42 |
| 4 | EfficientNetB1-4 | 0.55 | 62.5 | 62.8 | 61.1 | 62 | 64 |
| 5 | EfficientNetB1-5 | 0.601 | 48.8 | 48.5 | 37.4 | 42.2 | 128 |

model, which attained an accuracy of 69% with a batch size of 22. Its precision is 69%, recall 68% and recall 68.4%.

The above results show that the performance of the proposed BiLSTM+EfficientNetB1 using early fusion strategy is much higher than different text and image-based unimodal classification algorithms both in the case of multilingual (Urdu-English) tweets. Figure 6 is used to graphically represent the performance of different unimodal and BiLSTM+EfficientNetB1 classifiers.

### E. Addressing third research question (RQ3)

To address the third question of our research: How to evaluate the performance of the proposed BiLSTM+EfficientNetB1 system w.r.t to baseline multimodal multilingual HS detection techniques and different datasets? We have performed different experiments. We used three different models, namely BERT+ResNext (*Sai, Srivastava & Sharma, 2022*), LSTM+InceptionV3 (*Gomez et al., 2020*) and LSTM+InceptionResNetV2 (*Dwivedy & Roy, 2023*).

These models are primarily designed to perform the categorization of multimodal English tweets; however, we have used these models to classify multilingual (Urdu-English) Tweets. In addition, two multimodal transformer architectures, namely VisualBERT (*Lu et al., 2019*) and VilBERT (*Suryawanshi et al., 2020*), are used to classify multilingual multimodal tweets.

The results are presented in the following Table 12. The results indicate that the proposed early fusion-based DL BiLSTM+EfficientNetB1 classifier outperformed baseline models. Figure 7 is used to indicate the comparative performance of proposed early fusion-based models with baseline models.

To test the generalizability of our proposed model, we used state-of-the-art datasets. The details of the datasets are:

**Multimodal Hate Speech dataset (MMHS150K) (*Gomez et al., 2020*)**

A multimodal HS dataset for the English language with 150,000 tweets was created and named as MMHS150K (*Gomez et al., 2020*). The dataset consists of tweet text, image text (if any), and images.

**Multimodal Memes Dataset for Offensive Content (MultiOff) (*Suryawanshi et al., 2020*)**

This dataset contains 743 memes which are divided into offensive and non-offensive categories.

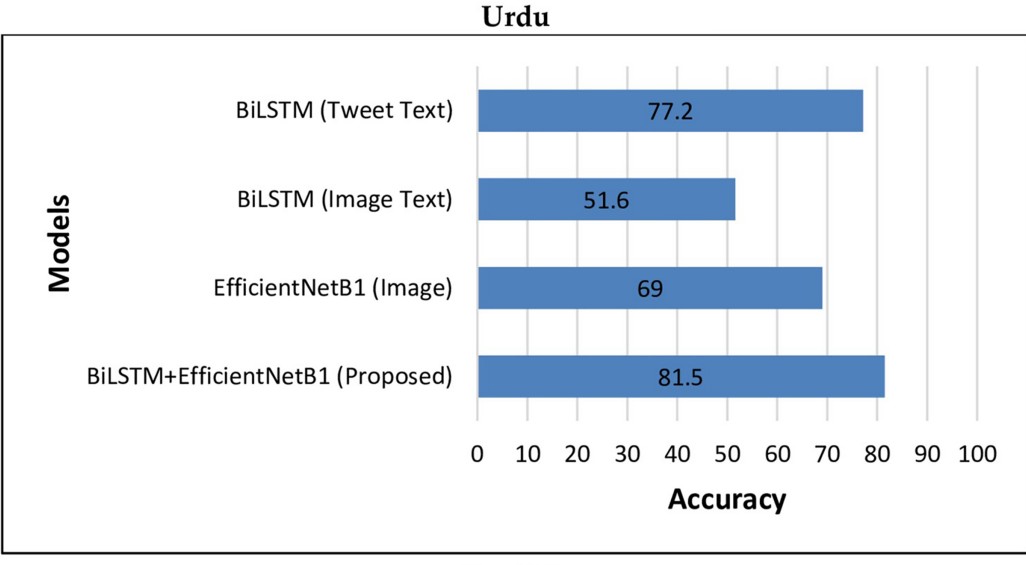

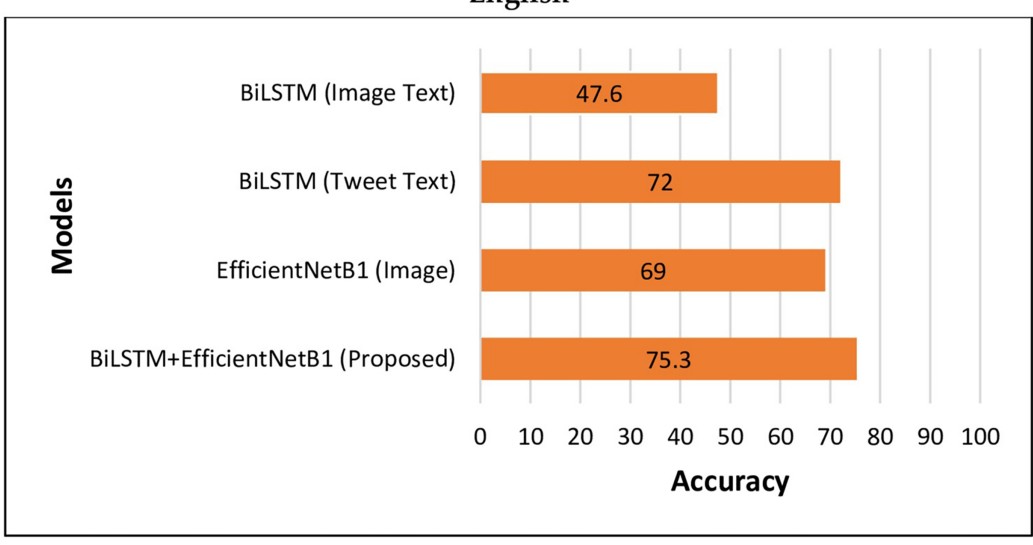

**Figure 6 Performance of BiLSTM+EfficientNetB1 *vs* unimodal models.**

### Hateful Memes Challenge (HMC) (*Kiela et al., 2018*)

Another dataset to recognize and classify hateful memes into corresponding classes. It contains 8,496 memes.

The results of various experiments are presented in Table 13. The results show that the proposed BiLSTM+EfficientNetB1 model outperformed comparing models on different datasets. This model attained accuracies of 69.01%, 88.5%, and 82.4% for the MultiOff, HMC, and MMHS150K dataset, respectively.

### F. Significance test (McNemar's test)

In this research, we have performed two different experiments to find out the statistical significance of the early fusion-based DL model, namely BiLSTM+EfficientNetB1, and the unimodal text-based classification model namely, BiLSTM. Now, to find out that the

**Table 12 Comparative performance of proposed & baseline early fusion-based models.**

| Model | Loss | Accuracy (%) | Precision (%) | Recall (%) | F1 (%) |
|---|---|---|---|---|---|
| Urdu | | | | | |
| VisualBERT (*Lu et al., 2019*) | 0.53 | 71.2 | 71 | 71.1 | 71.3 |
| VilBERT (*Suryawanshi et al., 2020*) | 0.65 | 39.1 | 38.8 | 38.8 | 38 |
| BERT+ResNext (*Sai, Srivastava & Sharma, 2022*) | 0.71 | 69 | 69 | 69 | 69 |
| LSTM+InceptionV3 (*Gomez et al., 2020*) | 0.49 | 73 | 73 | 73 | 73 |
| LSTM+InceptionResNetV2 (*Dwivedy & Roy, 2023*) | 0.51 | 76 | 74.1 | 75.9 | 74.9 |
| BiLSTM+EfficientNetB1 (Proposed) | 0.32 | 81.5 | 82.7 | 79.8 | 81.2 |
| English | | | | | |
| VisualBERT (*Lu et al., 2019*) | 0.63 | 63 | 63.5 | 63.4 | 63.5 |
| VilBERT (*Suryawanshi et al., 2020*) | 0.77 | 70.2 | 70.2 | 70.3 | 70.2 |
| BERT+ResNext (*Sai, Srivastava & Sharma, 2022*) | 0.45 | 58 | 58.8 | 58 | 58.8 |
| LSTM+InceptionV3 (*Gomez et al., 2020*) | 0.55 | 67 | 66.2 | 66.5 | 67 |
| LSTM+InceptionResNetV2 (*Dwivedy & Roy, 2023*) | 0.48 | 70 | 70 | 70 | 70 |
| BiLSTM+EfficientNetB1 (Proposed) | 0.4 | 75.3 | 75.5 | 75.5 | 75.5 |

performance of these two models is not obtained by chance and statistically distinct, we have selected 268 tweets using a random strategy. These tweets are provided as input to both classifiers. The results of predictions are presented in Table 14.

We have used McNemar's test to validate the null hypothesis. The null and alternate hypotheses are presented in the following:

$H_0$: The error rate of both classification models is the same.

$H_1$: The error rate of both classification models is significantly different.

Mathematically, the Chi-squared or McNemar's statistic is computed as:

$$\chi2 = \frac{(|b - c| - 1)^2}{(b + c)} \tag{21}$$

In the above equation, the $\chi^2$ is called Chi-squared statistic, the **b** and **c** are numbers which represent the discordant pair of samples. The number 1 represents the degree of freedom.

Table 15 is indicating the summarized score of different parameters. The Chi-squared statistic is 4.67, two-tailed **p-value** is **0.029**, and the degree of freedom is **1**.

### G. Discussion

The experiment performed to test the performance of unimodal text-based BiLSTM (Table 9, S#:2) classifiers showed poor performance with an accuracy of 77.2% in detecting HS from Urdu tweets. In another experiment, the proposed early fusion-based DL classifiers BiLSTM+EfficientNetB1 (Table 8, S#:2) outperformed with an accuracy of 81.5%.

After testing the significance of the two models, we uncovered that there is a significant difference between the BiLSTM model with text-only features and

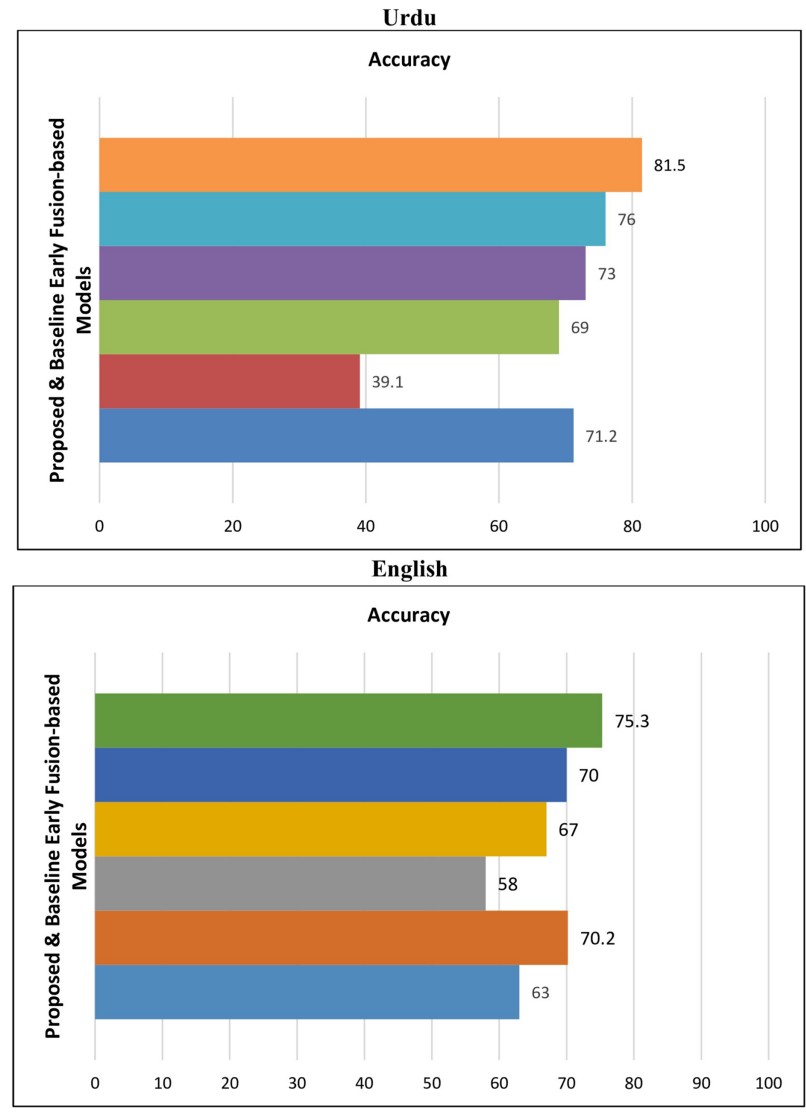

**Figure 7 Performance of proposed & baseline early fusion-based models.**

BiLSTM+EfficientNetB1 with both textual and visual features-based models. The results in Table 14 indicate that both models disagreed on 62 tweets (models processing features from different modalities behave differently during misclassification). To calculate the $p$-value, McNemar's test with continuity correction strategy was used. The chi-squared value is 4.67, and the two-tailed $p$-value is 0.029. We used one degree of freedom. Finally, the null hypothesis is rejected as $0.029 < 0.5$, and the alternate hypothesis is accepted.

Therefore, the proposed early fusion-based BiLSTM+EfficientNetB1 model using text and image features is statistically significant than the BiLSTM model with textual features. Hence, the statistical analysis shows that multimodal text and image features significantly enhance the performance of the proposed BiLSTM+EfficientNetB1 classifier. The proposed BiLSTM+EfficientNetB1 framework demonstrates its suitability for multimodal multilingual HS detection by addressing critical gaps in existing methods.

**Table 13 Comparative performance of proposed & existing models on different datasets.**

| Dataset | Model | Accuracy (%) | Precision (%) | Recall (%) | F1 (%) |
|---|---|---|---|---|---|
| MMHS150K | *Gomez et al. (2020)* | 68.5 | — | — | 70.4 |
| | *Chhabra & Vishwakarma (2024a)* | 80.8 | 71.08 | 73.8 | 72.4 |
| | Proposed | 82.4 | 80 | 81.5 | 81 |
| MultiOff | *Suryawanshi et al. (2020)* | — | 44 | 66 | 50 |
| | *Chhabra & Vishwakarma (2024a)* | 65.09 | 67.4 | 69.4 | 68.3 |
| | Proposed | 69.01 | 69.1 | 69.5 | 69.2 |
| HMC | *Kiela et al. (2018)* | 69.47 | — | — | — |
| | *Chhabra & Vishwakarma (2024a)* | 87.9 | 83.4 | 61.4 | 76.7 |
| | Proposed | 88.5 | 86 | 80 | 83.5 |

**Table 14 Result of BiLSTM & BiLSTM+EfficientNetB1 for significance testing.**

| BiLSTM+EfficientNetB1 (Proposed) | | BiLSTM | | Total |
|---|---|---|---|---|
| | | Correctly classified | Misclassified | |
| | Correctly classified | 187 | 40 | 227 |
| | Misclassified | 22 | 19 | 41 |
| | | 209 | 59 | 268 |

**Table 15 Summarized statistics of McNemar's test.**

| *P*-value | 0.029 |
|---|---|
| Statistic of McNemar's Chi-squared | 4.67 |
| Degrees of freedom | 1 |

The early fusion strategy integrates text and image features, enabling nuanced classification across modalities. Furthermore, the ablation study confirms the necessity of each component, such as BiLSTM for text processing and EfficientNetB1 for image analysis. Compared to baseline models, the proposed method consistently achieves superior performance, as evidenced by results in Table 12.

The robustness of the proposed BiLSTM+EfficientNetB1 framework lies in its capacity to integrate text and visual features through an early fusion strategy, which enhances its adaptability to diverse multimodal datasets. Extensive experiments demonstrate its superior performance not only on the MMHS11K dataset but also on established benchmarks such as MMHS150K and MultiOff. Furthermore, its ability to consistently outperform baseline unimodal and multimodal models highlights its resilience across various linguistic and visual complexities.

Statistical tests, including McNemar's test, validate its significant improvement over unimodal approaches, further underscoring its robustness. Future enhancements, such as

**Table 16 Abalation study.**

| Configuration | Accuracy | F1 score |
|---|---|---|
| Full model (BiLSTM+EfficientNetB1+Fusion) | 81.5% | 81.2% |
| Without BiLSTM (Text: CNN/LSTM) | 72.4% | 70.5% |
| Without EfficientNetB1 (Image: VGG19) | 74.3% | 73.0% |
| Text only (BiLSTM) | 76.8% | 75.5% |
| Image only (EfficientNetB1) | 70.9% | 69.8% |
| Without hyperparameter tuning | 77.1% | 76.4% |

leveraging transformer-based architectures, could further solidify this framework's generalizability and resilience.

While the BiLSTM and EfficientNetB1 models are established techniques, their application in a multimodal multilingual HS detection framework, particularly for Urdu-English tweets, is novel. Furthermore, our approach's competitive performance against state-of-the-art methods highlights its robustness. Future research could explore Transformer-based architectures like BERT or multimodal variants (*e.g.*, Vision Transformers) for further performance gains.

Similarly, generative adversarial networks (GANs) could be used to augment low-resource datasets, enhancing the model's ability to generalize across diverse data distributions. Furthermore, the proposed BiLSTM+EfficientNetB1 framework can be extended to form an explainable AI system by incorporating interpretability techniques.

For the textual modality, feature attribution methods such as SHAP (Shapley Additive Explanations) can be employed to identify the most influential words or phrases contributing to the classification of HS. Similarly, for the visual modality, Grad-CAM (gradient-weighted class activation mapping) can be utilized to generate heatmaps highlighting regions of the image that are most relevant to the model's predictions. By integrating these techniques, the framework can provide transparency into its decision-making process, enhancing trust and usability in real-world applications.

### H. Ablation study

We carried out a study to assess the impact of every element in the system. BiLSTM is used for text analysis and EfficientNetB1 for image processing. Along with the multimodal fusion approach, we employed to combine them all together smoothly and effectively. The outcomes are neatly presented in Table 16 below, which showcases how changes or eliminations of components affected the results.

The findings indicate that when either BiLSTM or EfficientNetB1 is omitted from the equation, there is a decrease in performance, which underscores their significance in the process of classification. Similarly, relying on either text or image individually diminishes the system's efficacy, highlighting the advantage of blending both modalities into a platform. Additionally, hyperparameter fine-tuning plays a role in enhancing the model's efficiency.

In summary, every module plays a pivotal role in ensuring top-notch results in detecting HS across modes of communication effectively and accurately. Combining text and image elements while tuning model settings greatly improves the system's precision and reliability.

**Algorithmatic complexity of the proposed system**

The complexity of the suggested system for detecting HS is influenced by three elements.

BiLSTM for text feature extraction: The bidirectional LSTM analyzes input sequences in Onwards and backward directions concurrently which leads to a time complexity of $O(2 * n * d^2)$ where n represents the length of the sequence and d stands for the dimension of the hidden state; this enables the model to comprehend contextual details from both preceding and subsequent states.

EfficientNetB1 for image feature extraction: EfficientNetB1 is recognized for being efficient in its use of parameters. It Operates with a complexity calculated as $O(k^2 * d^2 * m)$ where k represents the kernel size and d stands for depth while m signifies the size of the input image.

This assist in achieving computational efficiency with upto the mark accuracy.

Feature fusion and classification: The initial fusion of text and image features involves complexity level of $O(f_t + f_i)$ with $f_t$ representing the feature dimensions of text and $f_i$ representing those of images respectively. The classification layers introduce a complexity level of $O(n * m)$, where n and m denote the input and output neurons, in the connected layers.

The systems overall complexity is $O(2 * n * d^2 + d^2 * m)$ of two times n times d squared plus k.

Squared times d times m which demonstrates its effectiveness in managing types of information while delivering strong results.

## CONCLUSIONS AND FUTURE WORK

This study presents a novel DL framework for detecting multimodal HS in Urdu-English tweets, addressing the unique challenges posed by multilingual and multimodal content. By integrating BiLSTM and EfficientNetB1 models using an early fusion strategy, the proposed system effectively combines textual and visual features, achieving superior performance compared to unimodal and baseline multimodal approaches. The introduction of the MMHS11K dataset fills a critical gap in resources for low-resource languages, enabling future research in this domain. Although the study demonstrates promising outcomes, certain limitations persist, including the dataset size and the absence of advanced architectures such as Transformers. Future research could incorporate additional modalities, such as audio and video, or leverage generative and unsupervised learning techniques to enhance the framework's robustness and generalizability. This research underscores the importance of addressing multimodal and multilingual challenges in HS detection, paving the way for robust and inclusive solutions in online content moderation. Future work may incorporate Transformer-based models (*e.g.*, BERT, Vision Transformers) or GANs for data augmentation and improved feature extraction.

These advanced techniques could complement our current framework, further enhancing its novelty and performance.

## ACKNOWLEDGEMENTS

During the preparation of this work, the authors used an AI tool, namely Gemini, in order to correct grammatical mistakes and edit the language professionally. After using this tool/ service, the authors reviewed and edited the content as needed and take full responsibility for the content of the publication.

### Funding

This research is funded by the Information Security Research Group at King Abdulaziz University. The funders had no role in study design, data collection and analysis, decision to publish, or preparation of the manuscript.

### Grant Disclosures

The following grant information was disclosed by the authors:
King Abdulaziz University.

### Competing Interests

The authors declare that they have no competing interests.

### Author Contributions

- Furqan Khan Saddozai conceived and designed the experiments, performed the experiments, analyzed the data, performed the computation work, prepared figures and/or tables, authored or reviewed drafts of the article, and approved the final draft.
- Sahar K. Badri analyzed the data, authored or reviewed drafts of the article, and approved the final draft.
- Daniyal Alghazzawi performed the experiments, analyzed the data, authored or reviewed drafts of the article, and approved the final draft.
- Asad Khattak analyzed the data, prepared figures and/or tables, authored or reviewed drafts of the article, and approved the final draft.
- Muhammad Zubair Asghar conceived and designed the experiments, performed the experiments, analyzed the data, performed the computation work, prepared figures and/or tables, authored or reviewed drafts of the article, and approved the final draft.

### Ethics

The following information was supplied relating to ethical approvals (*i.e.*, approving body and any reference numbers):

The data used in this study was collected from publicly accessible online forums where users voluntarily share text-based content. No personally identifiable information (PII) was collected or utilized. The study complies with ethical guidelines on the use of publicly

available data, and additional consent from individual forum users was not required, as no private or restricted information was accessed.

## Data Availability

The dataset is available at GitHub, Zenodo, and figshare:

- https://github.com/FurqanSaddozai/Multimodal-Hate-Speech-Detection-A-Novel-Deep-Learning-Framework-for-Multilingual-Text-and-Images.

- Furqan Khan Saddozai, Sahar K Badri, M. Alghazzawi, D., Asad Khattak, & Zubair Asghar, M. (2025). FurqanSaddozai/Multimodal-Hate-Speech-Detection-A-Novel-Deep-Learning-Framework-for-Multilingual-Text-and-Images: v1.0.0 (v1.0.0) [Data set]. Zenodo. https://doi.org/10.5281/zenodo.14787594.

- SADDOZAI, FURQAN KHAN; Badri, Sahar K; Alghazzawi, Daniyal; Khattak, Asad; Asghar, Muhammad Zubair (2025). Multimodal Hate Speech Detection: A Novel Deep Learning Framework for Multilingual Text and Images. figshare. Dataset. https://doi.org/10.6084/m9.figshare.27310764.v3.

The tweets were accessed via the open-source snscrape toolkit, which allows for tweet retrieval without requiring API authentication:

Toolkit Reference: https://github.com/JustAnotherArchivist/snscrape.

API Documentation Reference: https://developer.x.com.

## Supplemental Information

Supplemental information for this article can be found online at http://dx.doi.org/10.7717/peerj-cs.2801#supplemental-information.

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
