# Peer review of "Multimodal hate speech detection: a novel deep learning framework for multilingual text and images"

_PeerJ Computer Science, doi:10.7717/peerj-cs.2801_

## Round 0.1 · original submission · Major Revisions

Both reviewers identified areas of improvement for this paper, among others needing more detail and justification on the methodologies used, both to back up the robustness and validity of the methodology, as well as to make it clearer to the reader and enable reproducibility.

While reviewers have provided specific references, authors are under no obligation to cite these specific references. You may consider expanding and updating the references accordingly, not necessarily using these references.

Please see reviewer comments who have provided details comments to address in a round of major revisions.

Reviewer 1 ·

Basic reporting

The manuscript presents an approach for detecting hate speech in multimodal Urdu-English tweets using a deep-learning framework that integrates BiLSTM and EfficientNetB1 models. This method effectively addresses the challenges of detecting hate speech in both text and image modalities, demonstrating improved performance over unimodal systems. However, the novelty of the work appears moderate, as it relies on well-established models like BiLSTM and EfficientNetB1. While these models offer good performance, the use of more advanced techniques, such as Transformer-based models or Generative Adversarial Networks (GANs), could potentially enhance the innovation and performance of the system.

The reliance on these established techniques, while effective, limits the work's originality as it builds on well-known methods without introducing a fundamentally new paradigm or algorithm.

Experimental design

the manuscript is well-presented and the methodology is well-justified. To enhance the quality of the manuscript , the following points should be addressed:

+ Avoid using the future tense. Maintain consistency by using the present tense in the third person and direct form. For example, replace phrases like "In this work, the following research question will be addressed: RQ1: How to categorize 95 ...." with "This study addresses: RQ1: How to categorize 95 ...."
+ Do not use abbreviations without prior definition, especially in the abstract, such as 'HS' and 'LSTM'

+ Replace informal connectors like 'so' with formal alternatives such as 'thus' or 'hence.'

+ Avoid non-scientific and unquantifiable words like: good, excellent, well....

+ Rewrite the abstract, introduction, and conclusion to ensure they are concise and consistent with academic writing standards.

+ The manuscript requires significant improvements in English language usage. Please review and revise the text to adopt a more formal tone.

+ I recommend adding the following recent (2024) and relevant bibliographic references:

• Malik, J. S., Qiao, H., Pang, G., & van den Hengel, A. (2024). Deep learning for hate speech detection: a comparative study. International Journal of Data Science and Analytics, 1-16.
• Mazari, A. C., Boudoukhani, N., & Djeffal, A. (2024). BERT-based ensemble learning for multi-aspect hate speech detection. Cluster Computing, 27(1), 325-339.
• Rawat, A., Kumar, S., & Samant, S. S. (2024). Hate speech detection in social media: Techniques, recent trends, and future challenges. Wiley Interdisciplinary Reviews: Computational Statistics, 16(2), e1648.

Validity of the findings

+ For the generalizability of the proposed method, multiple datasets should have been used for the experimentation.

Reviewer 2 ·

Basic reporting

The authors propose a novel deep-learning framework to detect multimodal hate speech from multilingual (Urdu-English) text and images. We collected and manually annotated a novel multimodal multilingual hate speech dataset, MMHS11K, for Urdu and English. We used BiLSTM and EfficientNetB1 models to extract textual and visual features. The BiLSTM model captures the context information from both forward and backward directions. On the other hand, EfficientNetB1 needs a smaller number of parameters and generally provides good results. Although the paper has good potential, it requires several changes as listed below:
1. Change the formatting based on the author's guidelines for the journal.
2. The abstract needs to be modified and reformatted in a more academic style.
3. The main contributions should be properly listed. It seems that the author has discussed all technical details as their contributions. It is recommended to list the most important or innovative points instead.
4. The motivation is confusing. The author should describe the shortcomings and strengths of this work compared to other similar methods in a separate section.
5. The language and formatting need improvement.
6. The equation numbers are not aligned.
7. Section E has a different line spacing than the rest of the manuscript.
8. Line 725 seems to be incomplete.
9. Section: Data Acquisition- the content is not justified and with different line spacing.
10. Improve the Literature Review by adding the following work on hate speech detection:
https://doi.org/10.1007/s00530-023-01051-8
https://doi.org/10.1016/j.engappai.2023.106991
arXiv preprint arXiv:2409.05136
arXiv preprint arXiv:2409.05134

11. The equations can be briefly described.
12. Can the authors use the proposed method to form an explainable AI? If yes, how?

Experimental design

1. Comparative analysis: What makes the proposed method suitable for this unique task? What new development to the proposed method have the authors added (compared to the existing approaches)? These points should be clarified.
2. Briefly discuss the robustness of the proposed model.

Validity of the findings

Overall, the paper is promising but requires the above-suggested changes before it can be accepted.

Additional comments

NA

---

## Round 0.2 · Minor Revisions

The paper has improved substantially and can be deemed acceptable in terms of scientific content. However, there are final suggestions from the reviewer to improve the writing quality to make it less informal, which I agree with. Please review the paper to improve the writing and to adhere to the standards of academic writing.

Reviewer 1 ·

Basic reporting

no comment

Experimental design

no comment

Validity of the findings

no comment

Additional comments

The authors have adequately addressed all the remarks from the previous version, except for my comments regarding the replacement of informal connectors like "so" and the avoidance of non-scientific, unquantifiable words such as "good." Therefore.

---

## Round 0.3 · accepted · Accept

I appreciate the authors' effort to address the minor revisions and to improve the writing to adhere to academic standards. I can now recommend acceptance.